# Demographics and risk of isolation due to sea level rise in the United States

Kelsea Best [1,2] ✉, Qian He[3,4], Allison C. Reilly [4], Deb A. Niemeier[4], Mitchell Anderson [5] & Tom Logan [5]

Within coastal communities, sea level rise (SLR) will result in widespread intermittent flooding and long-term inundation. Inundation effects will be evident, but isolation that arises from the loss of accessibility to critical services due to inundation of transportation networks may be less obvious. We examine who is most at risk of isolation due to SLR, which can inform community adaptation plans and help ensure that existing social vulnerabilities are not exacerbated. Combining socio-demographic data with an isolation metric, we identify social and economic disparities in risk of isolation under different SLR scenarios (1-10 ft) for the coastal U.S. We show that Black and Hispanic populations face a disproportionate risk of isolation at intermediate levels of SLR (4 ft and greater). Further, census tracts with higher rates of renters and older adults consistently face higher risk of isolation. These insights point to significant inequity in the burdens associated with SLR.

Disadvantaged populations, meaning those that have the fewest resources and least ability to adapt and respond, will experience the most severe effects of climate change[1–4]. Such disadvantaged groups often include racial minorities, older populations, relatively low-income populations, and renters. An estimated 20 million coastal residents in the U.S. will be at risk of inundation due to sea level rise (SLR) and/or storm surges by 2030[5], yet there is less evidence of how multiple and cascading burdens of SLR that are beyond direct inundation will affect disadvantaged populations. We argue that concentrating solely on adaptation to the inundation effects of SLR neglects more complex burdens of SLR, such as the isolation of communities and individuals from essential services, that may interact with social vulnerability to reinforce structures of inequality[6,7].

Unequal risk to SLR can manifest in different ways and across scales. In New York City, uneven exposure, inequitable adaptation responses, and historically discriminatory real estate development practices all act as potential drivers of disproportionate social vulnerability to flood risk across racial and ethnic groups[8]. Underinvestment in infrastructure is also prevalent in historically marginalized communities and results in a built environment less resilient to shocks such as natural hazards and the effects of climate change[9,10]. When considering the costs of adapting to flood risks, communities with relatively higher social disadvantage, including racial minorities, older populations, renters, and low-income communities may be more likely to be abandoned rather than protected[7]. In short, reduced capacity to adapt or retreat from natural hazards is often aligned with historical land use policies, less political power, lower degrees of collective efficacy, and existing racial and socioeconomic disadvantages[7,11,12].

Current literature relies almost exclusively on measures of inundation (direct flooding)[13–17] to assess community risk, yet the risk that a land parcel is inundated is by itself an insufficient measure of the burdens created by SLR. Consider how the inundation of non-residential land use and infrastructure, such as road networks, can disrupt access to, or isolate households from essential services such as hospitals, supermarkets, and schools[18–20]. Risk of isolation is a metric that can allow us to better characterize the short- and long-term effects of SLR effects on local communities.

[1]Department of Civil, Environmental and Geodetic Engineering, The Ohio State University, Columbus, OH, USA. [2]Knowlton School of Architecture, City and Regional Planning, The Ohio State University, Columbus, OH, USA. [3]Department of Geography, Planning, and Sustainability, Rowan University, Glassboro, NJ, USA. [4]Department of Civil and Environmental Engineering, University of Maryland, College Park, MD, USA. [5]Department of Civil and Natural Resources Engineering, University of Canterbury, Christchurch, New Zealand. ✉e-mail: best.309@osu.edu

In this paper, we ask two simple but fundamental questions: (1) How does risk of isolation vary among racial and ethnic groups along the coastal areas of the United States, and (2) How does the risk of isolation correlate with socio-demographic characteristics associated with social vulnerability (e.g., age, income, renter status, racial composition)? We argue that isolation creates a unique circumstance in which connectivity to essential services has been disrupted on a highly localized spatial scale[19,20]. In this paper, we measure risk of isolation using an established methodology that intersects Open-StreetMap (OSM) road network data with National Oceanographic and Atmospheric Administration (NOAA) mean higher high water (MHHW) scenarios for global SLR from 1 to 10 ft for all coastal counties in the continental U.S. Briefly, we compute the risk of isolation due to SLR by intersecting the U.S. OpenStreetMap (OSM) road network with NOAA's mean higher high water (MHHW) for global sea-level rise scenarios between 1 to 10 ft of global SLR at one-foot increments. Sea level will not rise uniformly. To account for this, we compute relative sea-level rise (RSLR) using tidal gauge data and SLR projections[21]. A census block is considered at risk of isolation if it lacks an available (non-inundated) route between its centroid and any fire stations or primary schools at MHHW. We consider these services essential, and they also serve as a proxy for other key service, civic, and activity areas.

Risk of isolation is calculated at the census block level, which is the smallest geographical unit for which U.S. census data is available. To estimate populations at risk of isolation, we use population data from the 2020 U.S. Census (most recent). We combine the risk of isolation with socioeconomic and demographic data (including race, median income, median age, and percent of renting households) from the American Community Survey (ACS) to assess the spatial distribution of risk burden due to a disruption of transport connectivity.

## Results
### Racial disparities in risk of isolation
Based on a distributive justice framework[22], we identify a disparity when the proportion of the group in the at-risk population is greater than the proportion of the group in the overall population. We focus this analysis on Black and Hispanic populations; these populations represent the largest racial minority groups in the U.S. We use "Black" to refer to populations that are Black or African American alone, "Hispanic" to refer to populations that are Hispanic or Latino, and "White" to refer to White alone, not Hispanic or Latino.

By aggregating all census block-level results and comparing them with the national percentage of each group, we find that Hispanic populations are overrepresented in the total population at risk of isolation at the SLR scenario beginning at 4 ft of SLR (Fig. 1), while Black populations are overrepresented after ~6 ft. White populations are underrepresented after 5 ft of SLR. When disparities emerge depends significantly on the SLR scenario pathway. When we compare two long-term SLR scenarios—an intermediate scenario (1.0 m of global SLR by 2100) and a high scenario (2.0 m of global SLR by 2100)[21]—we find strong evidence of disparate isolation effects in the intermediate scenario beginning by 2120 and the high scenario as early as 2090 (Fig. 2).

Aggregating census block-level risk of isolation results to the state level, we find that Black populations in Pennsylvania face a disproportionate risk of isolation at 3 ft of SLR relative to their representation in the state population (12.7% Black population in the state overall as of 2021). Hispanic populations in Florida, Louisiana, Mississippi, and Maine face a disproportionate risk of isolation at 3 ft of SLR relative to their representation in the state population (27.1%, 5.8%, 3.6%, and 2.1% Hispanic population in each state respectively).

Aggregating census block-level risk of isolation results to the county level, we find that the number of coastal counties where Black and Hispanic populations are disproportionately at risk of isolation relative to the overall county population increases as SLR increases (Fig. 3). Hispanic populations are especially vulnerable to the uneven risk of isolation. At SLR of 5 ft and 10 ft, 39.5% and more than 50%, respectively, of all coastal counties show disproportionate risk of isolation. Black populations do not fare much better. At 5 ft and 10 ft SLR, 24% and 30% of counties, respectively, show disproportionate risk of isolation for Black populations.

Black populations are disproportionately affected primarily in parts of the Northeast and California. We see counties where Hispanic populations are disproportionately affected across the study area, especially in Northern California, Louisiana, and large swaths of the Northeast (Fig. 4). In total, we estimate that 34 counties face a disproportionate risk of isolation for both Black and Hispanic populations at 3 ft of SLR. Fifteen of these 34 counties are in the Northeast and seven in California. There are six such counties in New Jersey alone (Supplementary Materials in Figs. S1 and S2). This suggests regional equity implications with uneven risk of isolation becoming more highly clustered as SLR increases.

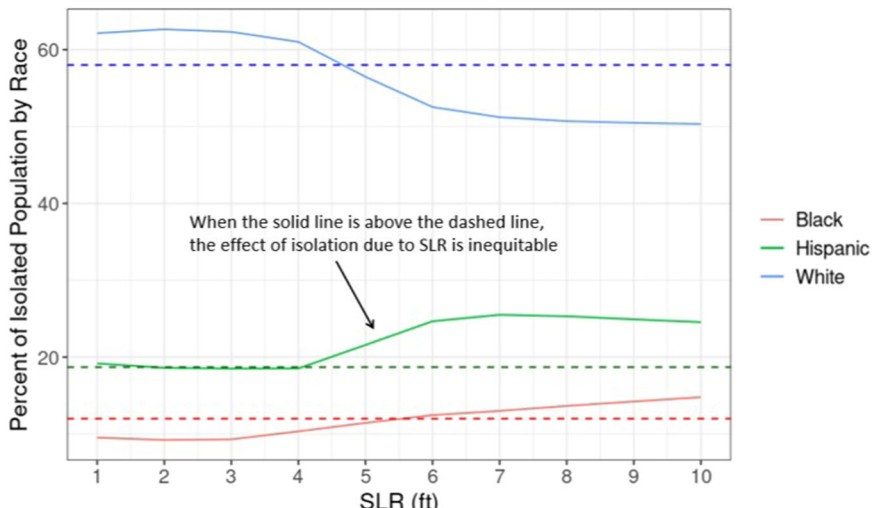

**Fig. 1 | Percent of population at risk of isolation by race at 1 to 10 ft of SLR.** Solid lines show the percentage of White (blue), Black (red), and Hispanic (green) populations at risk of isolation under each SLR scenario (1 to 10 ft). Dashed horizontal lines represent the percentage of that racial group in the overall U.S. population.

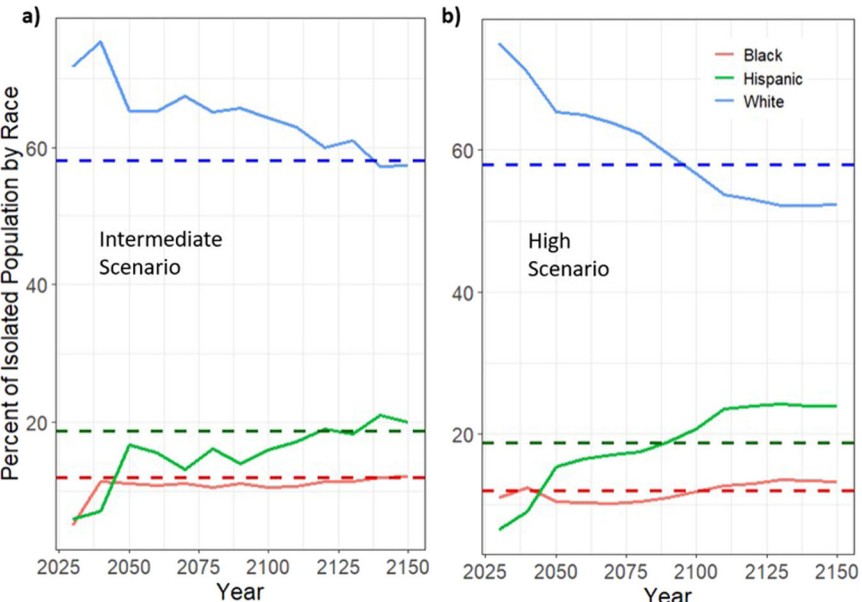

**Fig. 2 | Percent of isolated population by race per year at intermediate and high SLR scenarios.** Solid lines show the percentage of White (blue), Black (red), and Hispanic (green) populations at risk of isolation from 2030 to 2150 under intermediate (**a**) and high (**b**) SLR scenarios. Dashed horizontal lines represent the percentage of that racial group in the overall U.S. population. Therefore, a solid line above the dashed line suggests that that group is disproportionately affected by isolation.

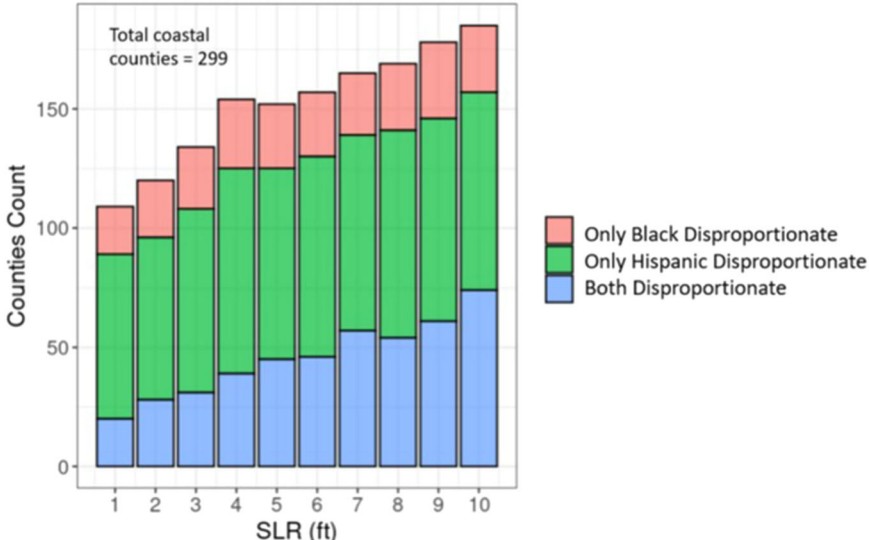

**Fig. 3 | Number of coastal counties where racial minorities are disproportionately at risk of isolation at 1 to 10 ft of SLR.** Numbers of coastal counties where it is predicted that only Black (red), only Hispanic (green), and both (blue) populations will be disproportionately at risk of isolation relative to the overall county population at varying levels of SLR.

To provide a localized perspective, we examine the degree to which racial minorities are disproportionately affected by isolation at the census tract level. We calculate the ratio of the percentage of the racial group in the population at risk of isolation to the percentage of the racial group in the overall county population. If this ratio is greater than 1, the racial group is disproportionately at risk of isolation. We find that the degree of disproportionate risk of isolation for racial minority populations is predicted to shift as SLR increases, consistent with results in Fig. 1. While there is a wide distribution, we see that Black and Hispanic populations in some tracts experience a rate of isolation more than ten times greater than their representation in the county population (Fig. 5). This points to the highly uneven and localized nature of the risk of isolation on these racial populations.

## Association between socio-demographic characteristics and risk of isolation

Racial minority status often interacts with other dimensions of social disadvantage, resulting in compounding and overlapping vulnerabilities[23,24]. To view these interactions, we compare the mean income, age, and percentage of White population across all census tracts included in the analysis for those tracts where the risk of isolation is predicted to disproportionately affect only Black populations, only Hispanic populations, both, and neither (Fig. 6). While we do not explicitly account for future demographic changes, a quick analysis of population shifts in our study area between 2009 and 2019 shows that income has remained largely unchanged (<2% average change), adjusting for inflation. Population age has increased (average 4.7% increase),

indicating that our exploration of the effects of risk of isolation on older populations is likely conservative. Similarly, the percentage of renters is increasing in these tracts (average 9%).

Tracts that are expected to have disproportionate risk of isolation for both Black and Hispanic minority populations also have the highest median household income (Fig. 6a). Across all levels of SLR, census tracts with the lowest percentage of the White population have Black

populations at disproportionate risk of isolation, while the areas with the highest percentage of White have disproportionate isolation of both Black and Hispanic populations (Fig. 6b). Finally, the oldest populations are expected to be in tracts with disproportionately Hispanic population at risk of isolation (at SLR = 1 ft). As SLR increases, tracts where both Hispanic and Black populations are disproportionately at risk of isolation are also the tracts with the oldest populations (Fig. 6c). These results highlight the compounded challenges and vulnerabilities experienced by populations at risk of isolation due to SLR.

Comparing characteristics of populations at risk of isolation and those at risk of direct inundation can provide insight into who may be left out of planning focused solely on inundation risk. To answer our second research question, we apply Generalized Linear Models (GLM) to three distinct outcome variables: the percentage of a census tract's population that is estimated to be at risk of isolation, the percentage of a census tract's population that is at risk of inundation, and the percentage of a tract's population that would be "missed" or left out in an analysis that relied solely on direct inundation as the measure of risk (Fig. 7 and Table 1). We define "missing" population as the difference between the population at risk of isolation and the population at risk of direct inundation. In these models, we include the level of SLR (1 to 10 ft) as an independent variable.

Tracts with older (higher median age), higher income, higher rates of minority, and higher proportions of renting populations are more likely to have a higher percentage of the population at risk of isolation. Findings are consistent for the risk of inundation, where age, percentage of the Black population, percentage of the Hispanic population, income, and percentage of renter households are positively correlated with percentage of the population at risk of inundation. Tracts with populations that are younger, have a higher median income, have more renters, and have lower percentages of Black and Hispanic populations are more likely to have a larger population that is missed from risk identification when only using the inundation metric (Fig. 7 and Table 1). Additional findings from individual scenario-based

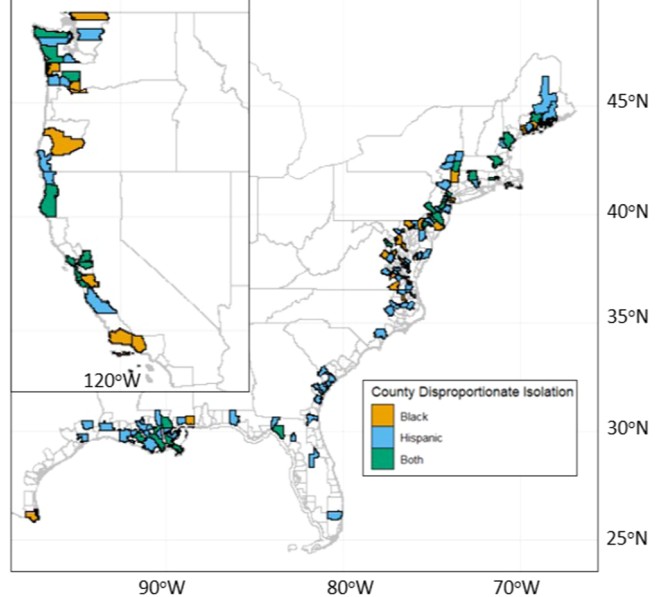

**Fig. 4 | Map of counties where 3 ft of SLR will disproportionately isolate racial minorities.** Counties where the SLR scenario of 3 ft is predicted to disproportionately result in isolation for Black (orange), Hispanic (blue), and both (green) populations relative to the overall county population.

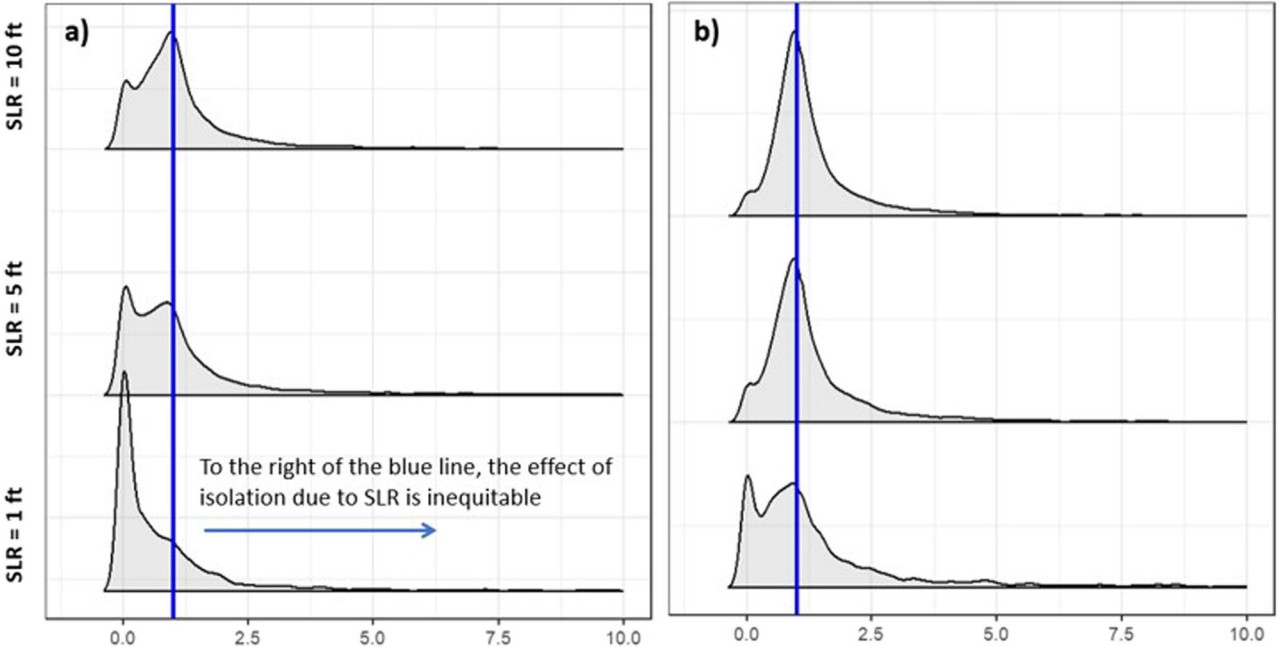

**Fig. 5 | Degree of the disproportionate effect of isolation on racial minorities.** Degree of the disproportionate effect for Black (**a**) and Hispanic (**b**) populations at the tract level. The blue line represents the line of equality ($x = 1$), meaning that any observations to the right of the blue line indicate disproportionate effects. Results are shown for SLR = 1, 5, and 10 ft.

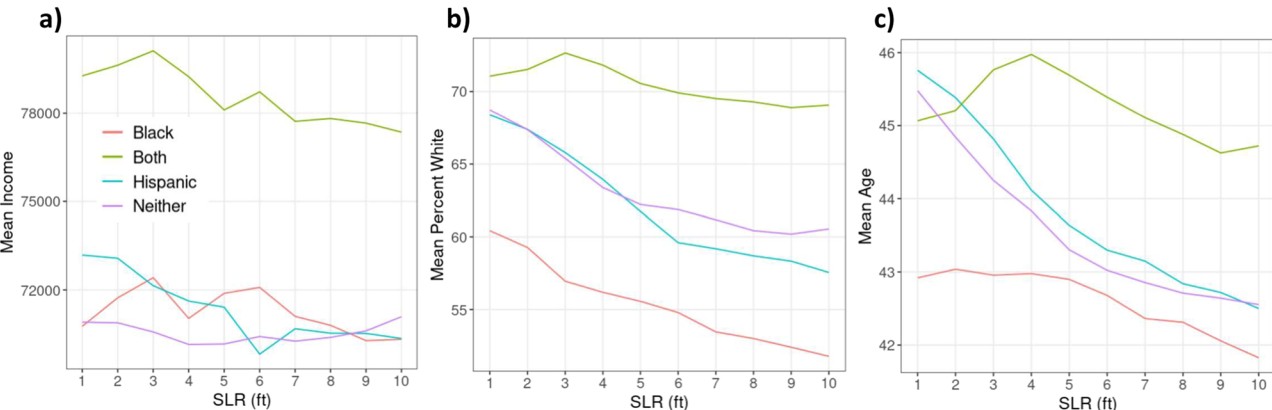

**Fig. 6 | Characteristics of tracts by disproportionate risk of isolation on racial minority populations.** Mean tract level income (**a**), percent White population (**b**), and age (**c**) for census tracts that are predicted to disproportionately result in the risk of isolation of Black population (red), Hispanic population (blue), both (green), and neither (purple) across varying levels of SLR.

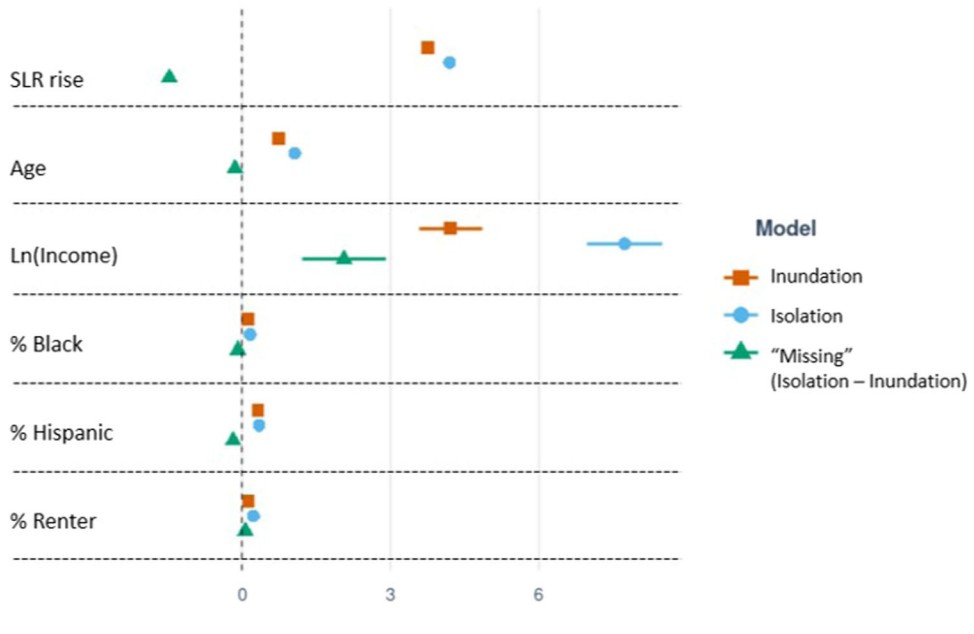

**Fig. 7 | Coefficient plots comparing isolation, inundation, and missing populations.** Coefficient plots comparing GLM outputs for isolation, inundation, and "missing" populations due to SLR. Center points represent the estimated coefficient, and bars represent ±standard errors. These coefficient plots correspond to the data presented in Table 1 (including estimates and standard errors).

Generalized Linear Models (GLM) regressions to predict the fraction of tract population at risk of isolation across 1 to 10 ft of sea level rise are available in Supplementary Materials (Fig. S3 and Table S1).

## Discussion

While the breadth of the isolation due to SLR is expected to be profound—as much as 30–90% more population is at risk compared to inundation and, in some cases, occurring decades early[17]—the characteristics of who is likely to be affected are not well documented. The question of who is burdened by SLR within a community and the ways that this burden might reinforce existing inequity is critical for equity and justice adaptation plans[3]. We examine the uneven burden of risk of isolation, demonstrating that historically marginalized populations such as Black and Hispanic populations are predicted to face a disproportionately greater risk of isolation, even at intermediate levels of SLR. In some tracts, racial minorities may be overrepresented in the population and at risk of isolation by a degree that is more than ten times greater than that group's representation in the overall population. The timing of these disproportionate burdens, however, depends heavily on future warming and SLR projections. The difference between an intermediate and high SLR scenario occurs multiple decades before Black and Hispanic populations are susceptible to overrepresentation in the population at risk of isolation when looking at the most aggregated level. This difference in timing reinforces the importance of mitigation to limit the degree of warming at a global scale in order to minimize the burdens of SLR on disadvantaged populations. However, the timing of risk of isolation on disadvantaged populations will vary at a very localized level, with some communities experiencing this risk significantly earlier than others.

Census tracts with older (higher median age) residents are more prone to risk of isolation. This reveals a new challenge of aging in place, especially when considering that older adults are facing decreased

**Table 1 | Results of GLM predicting percent of population at risk of isolation within a census tract**

| | % Isolation model | | | %Inundation model | | | % Missing model | | |
|---|---|---|---|---|---|---|---|---|---|
| | Coef. | St. Err. | $p > z$ | Coef. | St. Err. | $p > z$ | Coef. | St. Err. | $p > z$ |
| Age | 1.066 | 0.018 | <2e−16 | 0.741 | 0.015 | <2e−16 | −0.140 | 0.020 | <2e−16 |
| %Black | 0.164 | 0.009 | <2e−16 | 0.126 | 0.007 | <2e−16 | −0.082 | 0.010 | <2e−16 |
| %Hispanic | 0.345 | 0.008 | <2e−16 | 0.321 | 0.006 | <2e−16 | −0.178 | 0.008 | <2e−16 |
| Ln_Income | 7.729 | 0.389 | <2e−16 | 4.218 | 0.325 | <2e−16 | 2.062 | 0.431 | <2e−16 |
| %Renter | 0.232 | 0.009 | <2e−16 | 0.129 | 0.007 | <2e−16 | 0.068 | 0.009 | <2e−16 |
| SLR | 4.193 | 0.054 | <2e−16 | 3.751 | 0.045 | <2e−16 | −1.467 | 0.060 | <2e−16 |
| _cons | −141.00 | 4.763 | <2e−16 | −91.728 | 3.986 | <2e−16 | 33.046 | 5.286 | <2e−16 |
| Num. of Obs = 53,682 | | | | Num. of Obs = 53,682 | | | Num. of Obs = 53,272 | | |
| Adjusted $R^2$ = 0.173 | | | | Adjusted $R^2$ = 0.177 | | | Adjusted $R^2$ = 0.022 | | |

Output from GLM for SLR scenarios from 1 to 10 ft for percent population at risk of isolation in a census tract (%Isolation), percent population at risk of inundation (%Inundation), and percent of the impacted population that is not captured in the traditional inundation metric (%Missing). All p values are calculated using a two-sided t-test. Coefficients and standard errors are also plotted in Fig. 7.

physical capacity, greater barriers to personal mobility, and increased needs to access medical and healthcare facilities[25]. The increased risk of isolation for older adults is additionally concerning as older populations are less likely to move from high-risk areas due to limited physical and economic capacity, stronger aversion to relocation, or stronger ties to ancestral homes or neighborhoods[26–29]. Our findings not only underscore the vulnerability of older adults in coastal communities but also highlight the concern that this population group may experience further health consequences if they are unable to meet their needs in accessing essential destinations. Alternatively, health systems may need to adapt to reach these populations in place.

Census tracts with a greater percentage of population in renter status are also more likely to have a greater proportion of their population at risk of isolation. Renters face additional and unique challenges in responding to increased risk of isolation due to shortages in affordable rental housing and threats of eviction. According to the latest national dataset from EvictionLab[30,31], the mean eviction filing rate for U.S. coastal counties is 8.2% compared to the 4.6% national mean eviction filing rate in 2018; renters living in coastal counties face greater eviction risks. Renters are also more likely to be left out of programs designed to reduce exposure to future flooding and inundation, such as federal home-buyout programs[32,33].

The vulnerability of renters to isolation due to SLR is apparent in the investigation of "missing" populations left out of measurement in traditional inundation metrics. Tracts with higher rates of renters have a higher percent of "missing" population, which means that the effects of SLR are likely to be underestimated for these areas. Areas that are more likely to exhibit an underestimation of affected population (when only relying on inundation metrics compared to using isolation) are also higher proportion White population, higher income, and younger. This specific group might be considered more mobile in that they likely have the resources and capabilities to move from an area affected by SLR[34]. A younger population is also likely to have lower levels of place attachment which might otherwise work to keep them from moving, even as the effects of climate change increase[35]. It is important that these populations be captured in analysis and planning, especially when considering that the movement of these populations could contribute to shifts in residential profiles (including demographics and tax bases) across coastal communities.

This work has several important limitations. One limitation is that our analysis considers the entire population of a census block to be isolated based on transportation routes from the centroid of that census block. It is possible that additional inequities in the risk of isolation due to SLR exist at an even finer spatial scale, which we are currently not able to capture. This work also does not consider possible future updates to road networks or other adaptive

measures. There are also limitations in using census data for demographic information, as all census data has an associated margin of error. Therefore, there may be areas where populations, characteristics, and therefore risk of isolation are over- or under-estimated in this analysis.

As the impacts of global climate change increase, SLR is likely to affect where people live and move, resulting in population changes including potential shifts in racial and socioeconomic composition. Another limitation of this work is that we do not account for such demographic shifts over time. Still, understanding the risk of isolation for current populations is important for urban planners and policymakers to understand how climate impacts will reshape community characteristics and to create evidence-based adaptation plans. If accessibility challenges become too great, relocation may be a serious consideration for some residents, while others may be forced to remain in place due to a lack of resources[36,37]. Relocation is associated with unique implications for individual well-being such as physical and mental health[38] as well as regional housing availability and affordability[39,40]. Isolation due to SLR could interact with community characteristics by rendering certain communities more or less desirable. The movement of some households from areas at risk of isolation could contribute to the displacement of current residents in areas of in-migration[39,40]. In this way, isolation may contribute to additional burdens for residents even in nearby communities that are not directly affected. Future work should explore these complex dynamics of SLR, isolation, economic investment, and gentrification.

This study provides insight into how SLR could disproportionately isolate disadvantaged populations in the U.S. including racial and ethnic minorities, older populations, and renters. Based on these results, decision-makers should incorporate measures of risk of isolation within an explicit equity framework into climate adaptation planning and policies. The risk of isolation and associated uncertainties could provide additional risk measurements to inform Dynamic Adaptive Pathways Planning (DAPP) and long-range urban planning decisions. As the risk of isolation due to SLR could exacerbate existing socioeconomic disparities, it is critically important for future adaptation and planning policies to address the social and spatial disparity in climate vulnerability and ensure that the implementations are conducted in a just and equitable way.

## Methods
### Measuring risk of isolation
We draw upon the concept of risk of isolation in which isolation is identified when a census block has no un-inundated route available to access essential facilities (primary schools and fire stations) at a given SLR scenario[17]. We compute the risk of isolation due to SLR by intersecting the U.S. OpenStreetMap (OSM) road network with NOAA's

mean higher high water (MHHW) for global sea-level rise scenarios between 1 to 10 ft of global SLR at one-foot increments. The road network was obtained from OSM using the geofabrik.de API[41]. Location data for primary schools and fire stations (essential services considered in this analysis) were collected from the US Department of Homeland Security Homeland Infrastructure Foundation-Level Data (HIFLD) dataset for the U.S[42]. A census block is considered at risk of isolation if it lacks an available (non-inundated) route between its centroid and any fire stations or primary schools at MHHW. We consider these services essential, and they also serve as a proxy for other key services, civic, and activity areas. One limitation is that the estimates of future risk of isolation are based on current road networks and do not account for possible future changes to road infrastructure, such as elevating roadways or network expansion. Similarly, our analyses that consider socio-demographic characteristics use the most recent Census data (i.e., 2020) to investigate demographics but does not account for possible future shifts in population characteristics[43].

For timing of risk of isolation, we match each census block centroid to the nearest (by Euclidean distance) tidal gauge with relative sea-level projections from published climate scenarios and corresponding data[21]. We focus on the intermediate scenario (with a global mean sea level rise of 1.0 m by 2100) and high scenario (with a global mean sea level rise of 2.0 m by 2100). The intermediate scenario is considered the high end of likely SLR without rapid ice-sheet loss. The high projection represents the high end of SLR projections under high emissions. For both the intermediate and high scenarios, we consider the mean projections. A census block is considered isolated in a given 10-year interval (from 2030 to 2150) if the SLR projected at the nearest tidal gauge is equal to or greater than the SLR height needed to result in the isolation of that census block. The populations that are estimated to be at risk of isolation during a given 10-year interval are summed across all census blocks included in the analysis.

### Socio-demographic characteristics

For census block-level estimates of populations at risk of isolation, we use 2020 census data (most recent) as American Community Survey (ACS) data are not available at the block level. The census block is the smallest available geographic unit in the US census data. In 2020, the average census block contained 69.3 residents (with a standard deviation of 137.3) and 28.6 housing units (with a standard deviation of 62.7). 2020 Census data were retrieved using the National Historical Geographic Information System[44].

Based on the need of each analysis, we examine the contextual characteristics of populations at risk of isolation for all coastal counties in the continental U.S. at multiple geographical scales, including the tract (approximate population between 1200 and 8000), and county levels. For analyses greater than the census block level, we use the American Community Survey (ACS) 2019 (5-Year Estimates) demographic and socioeconomic characteristics data for our coastal counties and pull the data using the *tidycensus* package in R[45]. At the time of analysis, the 2020 census dataset has limitations and missing variables related to age, income, and renter status.

The ACS data include median household income, median age, households living in renter-occupied housing units, and data related to race. As mentioned, we focus this analysis on Black and Hispanic populations, as these populations represent the largest racial minority groups in the U.S. We use "Black" to refer to populations that are Black or African American alone, "Hispanic" to refer to populations that are Hispanic or Latino, and "White" to refer to White alone, not Hispanic or Latino in the ACS data.

We create a measure of "racial disparity isolation" for comparing the proportion of the racial/ethnic minority population (i.e., Black and Hispanic population) within the estimated population at risk of isolation to the proportion of that racial/ethnic minority population in the total area. In other words, if the proportion of a minority at risk of isolation is greater than the proportion of the minority in the overall population (at the tract or county level), then we consider the risk of isolation in that area to be disproportionately affecting the minority group. We identify and map locations where our approach predicts that racial/ethnic minorities will be disproportionately at risk of isolation at varying levels of SLR.

We analyze different spatial scales of risk of isolation (from block to national level) in order to explore how the risk and characteristics of those at risk vary by scale. Figures 1 and 2 present data that are aggregated for the entire continental U.S. The analyses presented in Figs. 3 and 4 then narrow the spatial scale to the county level. The remaining analyses use the tract level to assess in more detail the disproportionate risks of isolation for Black and Hispanic populations (Fig. 5) and how that risk of isolation interacts with other relevant socioeconomic characteristics including age, median income, and renter status (Figs. 6 and 7).

### Multivariate analysis

To further identify characteristics of census tracts susceptible to higher levels of isolation due to SLR as well as identify characteristics of populations that are not captured by analyses of inundation alone, we use our additional census data (i.e., age, income, and renters) along with isolation and inundation measurements for the multivariate regression analysis. We construct a dataset to examine the relationship between demographic and socioeconomic characteristics and the risk of isolation (percentage of population at risk of isolation per census tract, Eq. (1)) and the risk of inundation (percentage of population at risk of inundation per census tract, Eq. (2)). For these models, the population in a census block is considered exposed to inundation if the centroid of that block intersects with the MHHW extent[17]:

$$\%\text{Isolation} = \beta_0 + \beta_1 \text{SLR} + \beta_2 \text{MedianAge} + \beta_3 \text{Ln}(\text{MedianIncome}) + \beta_4 \%\text{Black} + \beta_5 \%\text{Hispanic} + \beta_6 \%\text{RenterHouseholds} \tag{1}$$

$$\%\text{Inundation} = \beta_0 + \beta_1 \text{SLR} + \beta_2 \text{MedianAge} + \beta_3 \text{Ln}(\text{MedianIncome}) + \beta_4 \%\text{Black} + \beta_5 \%\text{Hispanic} + \beta_6 \%\text{RenterHouseholds} \tag{2}$$

We also investigate the characteristics of the population "missing" from analyses that depend solely on inundation risk measures (Eq. (3)). The percent "missing" population is calculated as the estimated population at risk of isolation minus the estimated population at risk of inundation and divided by the population at risk of isolation at each census tract and for each level of SLR. The multivariate regression analysis allows us to further estimate how socioeconomic variables of interest are correlated with SLR burdens:

$$\%\text{Missing} = \beta_0 + \beta_1 \text{SLR} + \beta_2 \text{MedianAge} + \beta_3 \text{Ln}(\text{MedianIncome}) + \beta_4 \%\text{Black} + \beta_5 \%\text{Hispanic} + \beta_6 \%\text{RenterHouseholds} \tag{3}$$

### Reporting summary

Further information on research design is available in the Nature Portfolio Reporting Summary linked to this article.

## Data availability

The data generated for population at risk of isolation are presented on our interactive dashboard https://research.urbanintelligence.co.nz/slr-usa. The remaining data are publicly available and detailed in "Methods".

## Code availability

Code for estimating the risk of isolation metric is available at https://github.com/urutau-nz/usa_slr.

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

## Acknowledgements

The work was partially funded by the Clark Distinguished Chair Endowment (D.A.N.). This research was also supported by the National Science Foundation (grant nos. 1940273 and 2145509) (A.C.R.). Any opinions, findings, conclusions or recommendations presented in this paper are those of the authors and do not necessarily reflect the views of the National Science Foundation. Partial funding for open access provided by the UMD Libraries' Open Access Publishing Fund. The support of the sponsors is gratefully acknowledged. We also thank the University of Canterbury's Cluster for Community and Urban Resilience (CURe) and the Department of Civil and Natural Resources Engineering.

## Author contributions

K.B. was involved in conceptualization, methodology, analysis, visualization, writing the original draft, and reviewing and editing. Q.H. was involved in conceptualization, methodology, analysis, writing the original draft, and reviewing and editing. A.C.R. and D.A.N. were involved with conceptualization, reviewing, and editing. M.A. and T.L. were involved in initial methodology, software, data curation, reviewing and editing.

## Competing interests

T.L. and M.A. have shares in the risk consulting firm Urban Intelligence. All other authors declare no competing interests.
