## [Peer Review File · Nature Communications]

Demographics and risk of isolation due to sea level rise in the United StatesREVIEWER COMMENTS

Reviewer #1 (Remarks to the Author):

This study evaluates how sea level rise will disproportionately impact vulnerable populations by using an isolation metric with geospatial sea level rise inundation and socioeconomic data. The analyses build on previous work by Logan et al. 2023 to develop the isolation metric and has the potential to provide valuable information on how sea level rise impacts on networked infrastructure will create additional societal impacts that are not represented using conventional assessments of property inundation. Given serious race and ethnicity-based disparities in coastal flood risk identified in previous research (e.g. Hardy et al. 2018, Herreros-Cantis 2020, Handwerger et al. 2021), this study provides some potentially noteworthy results, including that:

- Census tracts with older, higher income, higher rates of minority, and higher proportions of renting populations are more likely to become isolated
- Communities with older residents are more prone to isolation risk. This reveals a new challenge of aging in place, especially when considering that older adults are facing decreased physical capacity, greater barriers to personal mobility, and increased needs to access medical and healthcare facilities.

The motivation, importance and potential implications for this study are very well-written in the current manuscript.

However, this study does not provide sufficient detail on the study methods for readers to fully interpret the study results and/or to reproduce the work, specifically:

- Demographic data are not available at the Block-level (only Block Group) through the American Community Survey so it is not possible for the study to have been conducted as the authors described at the Block scale. Did the Block-level analysis of total population and race/ethnicity actually come from the 2020 Decennial Census? Or were the analyses actually conducted at the Block Group and not the Block-level as it says throughout the paper? The study needs to be clear about what analyses were conducted at what geographic scales.
- The authors need to provide basic information on how vulnerable demographics were classified from the Census data. The US Decennial Census/American Community Survey provides data on both race and Hispanic/Non-Hispanic ethnicity of populations. Are the 'Black' and 'White' populations exposed here exclusively non-Hispanic? Does 'Hispanic' here include people that identify as being of Hispanic ethnicity that are of any race? How are people of other races or that identify as being of more than one race considered? How were 'older' populations identified? Which income variable was used (there are many provided in the American Community Survey)? These are essential details that need to be provided in the Online Methods for any analyses based on US Census data.

There are also some issues with the paper organization:

- While the authors can reference the Logan et al. 2023 paper for the details of the isolation analyses, they should provide a basic definition of what the isolation metric is. The main text of the paper only provides the vague description of the metric as "an established methodology that intersects OpenStreetMap (OSM) road network data with National Oceanographic and Atmospheric Administration

(NOAA) mean higher high water (MHHW) scenarios", but there's no way for the reader to otherwise know what they mean by 'isolation' without referring to the Online Methods.

- The authors use the descriptor 'missing' in the Discussion section of the main text (Page 11 line 217), but need to define what they mean by 'missing' here without the reader having to refer to the Online Methods.

- The Online Methods must more clearly describe which socio-economic analyses were conducted at each scale so that the reader can link the analyses with the results in the main text. In the current manuscript, it's very difficult to follow what analyses were conducted at each scale (e.g. county, tract). The methods only provide that characteristics were assessed "based on the need of each analyses" (page 13 line 279), which seems arbitrary as written.

In addition, some additional analyses are also needed to provide context for, and better support, the study's conclusions and claims:

- The study evaluates the disparities amongst Census Block-level populations within counties and census tracts, but these aren't the most useful geographies to support decision-making or to understand disparities in vulnerability. Instead, it would be helpful to provide an additional assessment of how/if vulnerable populations in each state, region, and the U.S. as a whole will be disproportionately burdened by isolation with sea level rise. This type of aggregation of the study results would be very straightforward to perform and would provide more support for the study's broader conclusions.

- The study should better explain why other vulnerable racial groups in the United States (e.g. Asian and Native American) were excluded from the assessment.

- The study analyses seem to completely exclude the states of Hawai'i and Alaska, along with U.S. Territories with high populations of vulnerable populations such as Puerto Rico and The Virgin Islands. The study should be more clear that it only evaluates vulnerable populations in the Continental United States.

Cited references:

Hardy, R.D. and Hauer, M.E., 2018. Social vulnerability projections improve sea-level rise risk assessments. *Applied Geography*, 91, pp.10-20.

Handwerger, L.R., Sugg, M.M. and Runkle, J.D., 2021. Present and future sea level rise at the intersection of race and poverty in the Carolinas: A geospatial analysis. *The Journal of Climate Change and Health*, 3, p.100028.

Herreros-Cantis, P., Olivotto, V., Grabowski, Z.J. and McPhearson, T., 2020. Shifting landscapes of coastal flood risk: environmental (in) justice of urban change, sea level rise, and differential vulnerability in New York City. *Urban transformations*, 2(1), pp.1-28.

Logan, T.M., Anderson, M.J. and Reilly, A.C., 2023. Risk of isolation increases the expected burden from sea-level rise. *Nature Climate Change*, 13(4), pp.397-402.

Reviewer #2 (Remarks to the Author):

The authors present an analysis of the socio-demographic characteristics of the population at risk of "isolation" from sea level rise, a measure that reflects whether there is an unflooded path from the centroid of a census block to a fire station and primary school at mean higher high water. They show that the populations at risk of isolation are disproportionately White from 1-4ft of sea level rise, after which the population at risk of isolation becomes disproportionately Black and Hispanic. They also analyze differences related to income, age, and renter/homeowner status.

The manuscript seeks to address an important question about who is affected by sea level rise and over what time frame. The focus on isolation, rather than housing or land inundation, distinguishes this analysis from others similar to it. My main concerns revolve around the clarity of the methods – I didn't understand the methods well enough to assess the findings, lack of discussion of limitations, and inconsistent wording.

Major comments:

I am assuming that Figure 1 and Figure 2 are based on summing up the populations at the block level and then comparing against the nation and the county population, but that could be more clear. The Supplementary refers to block groups, but that doesn't appear anywhere in the main text. The Methods describes socio-demographic characteristics from the ACS, but the ACS doesn't report any measures at the block level as far as I am aware (I think the block group is the lowest level for the ACS, so the block data would come from the Census). It is not clear to me why the tract-county comparison (lines 116-123) are "localized" if the preceding section is about the block-county comparison, but maybe I'm misunderstanding the preceding section.

The main text does not contain enough info on methods (I realize the methods come separately, but it's so minimal in the main text that I stopped reading, scrolled to the methods, and returned). I think a few more sentences would at least provide enough detail for the main text to stand on its own, while obviously readers more interested in the details can go to the methods section specifically. For example, the main text doesn't mention that the measure of isolation is done at the census block scale, using the block centroid.

For Figure 6 and associated results: the equation for the multivariate modeling is not included anywhere, and it was not clear to me what the units of SLR are and how it relates to the outcome variable – do you run the model for multiple levels of SLR and thus multiple outcome measures of inundation, isolation, etc.? The paper states that tracts with older, higher income, etc. are more likely to become isolated, but isn't it actually showing that a higher share of the tract is at risk of isolation, since the tract isn't being classified as isolated/not isolated?

Second, there were a few major limitations that I did not see addressed anywhere in the text. Modeling isolation based on a census block centroid makes sense in urban areas where blocks are very small, but could be more substantially off for larger (in area) census blocks. Elevation can change substantially within some census blocks (thus much wider range of actual isolation) compared to others. The results assume that the entire population of the block is either isolated or not, and thus do not account for the within-block distribution of populations – it could be that socially vulnerable populations live in the most flood-prone parts of the block, so these results are an underestimate, or vice versa.

Minor comments:

- Overall: "risk of isolation," "isolated," "isolation risk exposure," "effect of isolation due to SLR" are all used seemingly interchangeably throughout the paper. Is this intentional or are there subtle differences in the meanings? Suggest being consistent with how you describe what is being measured.
- Abstract: specify the SLR amounts rather than "intermediate" which is subjective

- Page 1, Lines 38-48 (and in some other parts of the paper): use of the terms hazard, risk, exposure, and vulnerability is inconsistent – need to either state definitions or be consistent with existing ones, e.g. IPCC (hazard, exposure, vulnerability as components of risk).
- Page 1, Lines 43-46: this is a modeled result using assumptions about economically optimal adaptation – this is not an empirical finding about likelihood of abandonment or protection.
- Page 1, Lines 49-54: the distinction being drawn in this paper is not about inundation vs. some other metric of exposure, it's about what is being inundated – a parcel, a % of land, vs. the roadways needed to access critical services.
- Figure 4: while there are some tracts where there are high ratios, it looks like in the majority of tracts, Hispanic and Black populations are not disproportionately at risk of isolation? I suggest avoiding cherry picking of specific results to describe here. Also flagging “disproportionate impact” as I don't know if “impact” is the right term to use here, and it might be clearer to actually put the ratio in the axis label instead.
- Does Figure 5 only include tracts that are isolated or where at least one of the blocks/block groups within it are isolated?
- There were a few places where the language was confusing and hard to understand. For example, Lines 134-135: “...for those tracts that are predicted to disproportionately affect only...” it implies the tracts are affecting something. Similarly, Lines 107-108: “Both Black and Hispanic populations are estimated to result in disproportionate isolation...”

Reviewer #3 (Remarks to the Author):

The authors examine how an often-overlooked artifact of sea level rise, isolation, and its consequences will effect the accessibility to critical services due to inundation of transportation networks. The study utilizes several socio-demographic datasets in combination with sea level rise data from NOAA. Overall I think the manuscript will be of interest to planners but there are a couple clarifications/edits needed in regards to the sea level terminology and datasets.

1) Inundation vs. flooding needs to be clarified and consistently used throughout the manuscript. Inundation should be considered any need at or below mean higher high water (MHHW) which is the agreement (for now) defining typically dry from typically wet land often referred to as 'ground level' by NOAA National Weather Service.

2) Related, why should intersection of MHHW be considered the time horizon for 'isolation'. MHHW has a definition of being exceeded about every other day. Though this could be argued as an effective limit, where is the justification? This criteria needs to be introduced, discussed and better justified. One could argue isolation would occur prior to the intersection with MHHW (numerically approximated to about 180 days/year). How about 30 or 60 days per year? Perhaps the transportation network itself becomes compromised long before this MHHW intersection occurs. Again, if the goal is providing realistic future estimates of 'isolation' this criteria needs serious consideration. The authors could use the extreme water level probabilities to estimate future high water or flood frequencies to refine the isolation time horizons.

3) The 1-10 foot contours on NOAA SEa Level Rise Viewer are not themselves scenarios of sea level rise (SLR). The most recent scenarios are from the 2022 Interagency report (Sweet et al., 2022) which is appropriately referenced in the manuscript. The authors state that they use the SLR scenarios from the tide gauges, but the 2022 report also provided 1-degree gridded SLR scenarios which might better resolve spatially. If the goal is to be as realistic as possible, this gridded dataset should be used. Also, though the 2022 scenarios are defined as Low, Int Low, Intermediate, Int High and High, if 'more likely' estimates are sought, more emphasis should be given to the Int Low and Intermediate scenarios. The High is the upper end of plausible rise by 2100 and not very likely to occur.

4) It is unclear how the time horizons for isolation under the Intermediate and High SLR scenarios were estimated and presented as one number for the country. It is important to attach time horizons to the SLR scenarios, not simply state, e.g. 'with 5' or 10' of sea level rise'. By when, where?

REVIEWER COMMENTS

Reviewer #1 (Remarks to the Author):

This study evaluates how sea level rise will disproportionately impact vulnerable populations by using an isolation metric with geospatial sea level rise inundation and socioeconomic data. The analyses build on previous work by Logan et al. 2023 to develop the isolation metric and has the potential to provide valuable information on how sea level rise impacts on networked infrastructure will create additional societal impacts that are not represented using conventional assessments of property inundation. Given serious race and ethnicity-based disparities in coastal flood risk identified in previous research (e.g. Hardy et al. 2018, Herreros-Cantis 2020, Handwerger et al. 2021), this study provides some potentially noteworthy results, including that:

- Census tracts with older, higher income, higher rates of minority, and higher proportions of renting populations are more likely to become isolated
- Communities with older residents are more prone to isolation risk. This reveals a new challenge of aging in place, especially when considering that older adults are facing decreased physical capacity, greater barriers to personal mobility, and increased needs to access medical and healthcare facilities.

The motivation, importance and potential implications for this study are very well-written in the current manuscript.

We thank the reviewer for these encouraging comments.

However, this study does not provide sufficient detail on the study methods for readers to fully interpret the study results and/or to reproduce the work, specifically:

- Demographic data are not available at the Block-level (only Block Group) through the American Community Survey so it is not possible for the study to have been conducted as the authors described at the Block scale. Did the Block-level analysis of total population and race/ethnicity actually come from the 2020 Decennial Census? Or were the analyses actually conducted at the Block Group and not the Block-level as it says throughout the paper? The study needs to be clear about what analyses were conducted at what geographic scales.

We apologize for the confusion here. We have clarified in the main body of the text and in the Methods section that the risk of isolation is calculated at the block level using 2020 Decennial Census data. The other socioeconomic characteristics for the analyses aggregated to larger spatial scales use the ACS data. We have added much more detail to the main body and Methods to guide the reader.

- The authors need to provide basic information on how vulnerable demographics were

classified from the Census data. The US Decennial Census/American Community Survey provides data on both race and Hispanic/Non-Hispanic ethnicity of populations. Are the 'Black' and 'White' populations exposed here exclusively non-Hispanic? Does 'Hispanic' here include people that identify as being of Hispanic ethnicity that are of any race? How are people of other races or that identify as being of more than one race considered? How were 'older' populations identified? Which income variable was used (there are many provided in the American Community Survey)? These are essential details that need to be provided in the Online Methods for any analyses based on US Census data.

Thank you. We have added the following to the main body of the text:

We focus this analysis on Black and Hispanic populations, as these groups represent the largest racial minority groups in the U.S. We use “Black” to refer to populations that are Black or African American alone, “Hispanic” to refer to populations that are Hispanic or Latino, and “White” to refer to White alone, not Hispanic or Latino.

We have also clarified in the main body that “older” simply refers to higher median age.

In the methods section, we clarify that income data is median household income. Age data is median age. Renter data are the standard census definition “households living in renter-occupied units”.

There are also some issues with the paper organization:

- While the authors can reference the Logan et al. 2023 paper for the details of the isolation analyses, they should provide a basic definition of what the isolation metric is. The main text of the paper only provides the vague description of the metric as "an established methodology that intersects OpenStreetMap (OSM) road network data with National Oceanographic and Atmospheric Administration (NOAA) mean higher high water (MHHW) scenarios", but there's no way for the reader to otherwise know what they mean by 'isolation' without referring to the Online Methods.

This is a good point. We have clarified in the main text that the isolation metric refers to:

Briefly, we compute the risk of isolation due to SLR by intersecting the U.S. OpenStreetMap (OSM) road network with NOAA’s mean higher high water (MHHW) for global sea-level rise scenarios between 1 to 10 ft of global SLR at one-foot increments. Sea level will not rise uniformly. To account for this, we compute relative sea-level rise (RSLR) using tidal gauge data and SLR projections from Sweet et al. (2022). A census block is considered at risk of isolation if it lacks an available (non-inundated) route between its centroid and any fire stations or primary schools at MHHW. We consider these services essential, and they also serve as a proxy for other key service, civic, and activity areas. The data, methods, and assumptions for calculating risk of isolation as well as limitations can be found in Logan et al. (2023).

- The authors use the descriptor 'missing' in the Discussion section of the main text (Page 11 line 217), but need to define what they mean by 'missing' here without the reader having to refer to the Online Methods.

We have clarified on page 11 where the reviewer points out : “We define “missing” population as the difference between the population at risk of isolation and the population at risk of direct inundation.”

- The Online Methods must more clearly describe which socio-economic analyses were conducted at each scale so that the reader can link the analyses with the results in the main text. In the current manuscript, it's very difficult to follow what analyses were conducted at each scale (e.g. county, tract). The methods only provide that characteristics were assessed "based on the need of each analyses" (page 13 line 279), which seems arbitrary as written.

We see the reviewer’s point that it is difficult to follow the scale of analysis as the manuscript was written. We have added the following to our Methods section to guide the reader:

We analyze different spatial scales of risk of isolation (from block to national level) in order to explore how the risk and characteristics of those at risk vary by scale. Figures 1 and 2 present data that is aggregated for the entire continental U.S. The analyses presented in Figures 3 and 4 then narrow the spatial scale to the county level. The remaining analyses use the tract level to assess in more detail the disproportionate risks of isolation for Black and Hispanic populations (Figure 5) and how that risk of isolation interacts with other relevant socioeconomic characteristics including age, median income, and renter status (Figures 6, 7).

In addition, some additional analyses are also needed to provide context for, and better support, the study's conclusions and claims:

- The study evaluates the disparities amongst Census Block-level populations within counties and census tracts, but these aren't the most useful geographies to support decision-making or to understand disparities in vulnerability. Instead, it would be helpful to provide an additional assessment of how/if vulnerable populations in each state, region, and the U.S. as a whole will be disproportionately burdened by isolation with sea level rise. This type of aggregation of the study results would be very straightforward to perform and would provide more support for the study's broader conclusions.

We thank the reviewer for this comment. Taking the reviewer’s suggestion, we have conducted a state-level analysis which we now include. For this state level analysis, we identify the states where 3 ft of SLR would result in a disproportionate risk of isolation for Black and Hispanic populations (similar to Figures S1 and S2 but this time at the state level). We write:

If we aggregate census block-level risk of isolation results to the state level, we find that Black populations in Pennsylvania are predicted to face a disproportionate risk of isolation at 3 ft of SLR relative to their representation in the state population (12.7% as of 2021). Hispanic populations in Florida, Louisiana, Mississippi, and Maine are predicted to face a disproportionate risk of isolation at 3 ft of SLR relative to their representation in the state population (27.1%, 5.8%, 3.6%, and 2.1% respectively).

- The study should better explain why other vulnerable racial groups in the United States (e.g. Asian and Native American) were excluded from the assessment.

We have added the following explanation: We focus this analysis on Black and Hispanic populations, as these groups represent the largest racial minority groups in the U.S. We use “Black” to refer to populations that are Black or African American alone, “Hispanic” to refer to populations that are Hispanic or Latino, and “White” to refer to White alone, not Hispanic or Latino.

We note also that traditionally larger Black and Hispanic communities have resided in coastal communities in California and much of the southeast.

- The study analyses seem to completely exclude the states of Hawai'i and Alaska, along with U.S. Territories with high populations of vulnerable populations such as Puerto Rico and The Virgin Islands. The study should be more clear that it only evaluates vulnerable populations in the Continental United States.

We have clarified that our analysis focuses on the continental U.S.

Cited references:

Hardy, R.D. and Hauer, M.E., 2018. Social vulnerability projections improve sea-level rise risk assessments. *Applied Geography*, 91, pp.10-20.

Handwerger, L.R., Sugg, M.M. and Runkle, J.D., 2021. Present and future sea level rise at the intersection of race and poverty in the Carolinas: A geospatial analysis. *The Journal of Climate Change and Health*, 3, p.100028.

Herreros-Cantis, P., Olivotto, V., Grabowski, Z.J. and McPhearson, T., 2020. Shifting landscapes of coastal flood risk: environmental (in) justice of urban change, sea level rise, and differential vulnerability in New York City. *Urban transformations*, 2(1), pp.1-28.

Logan, T.M., Anderson, M.J. and Reilly, A.C., 2023. Risk of isolation increases the expected burden from sea-level rise. *Nature Climate Change*, 13(4), pp.397-402.

Reviewer #2 (Remarks to the Author):

The authors present an analysis of the socio-demographic characteristics of the population at risk of “isolation” from sea level rise, a measure that reflects whether there is an unflooded path from the centroid of a census block to a fire station and primary school at mean higher high water. They show that the populations at risk of isolation are disproportionately White from 1-4ft of sea level rise, after which the population at risk of isolation becomes disproportionately Black and Hispanic. They also analyze differences related to income, age, and renter/homeowner status.

The manuscript seeks to address an important question about who is affected by sea level rise and over what time frame. The focus on isolation, rather than housing or land inundation,

distinguishes this analysis from others similar to it. My main concerns revolve around the clarity of the methods – I didn't understand the methods well enough to assess the findings, lack of discussion of limitations, and inconsistent wording.

We thank the reviewer for the helpful comments. We have taken steps to improve the clarity of our methods.

Major comments:

I am assuming that Figure 1 and Figure 2 are based on summing up the populations at the block level and then comparing against the nation and the county population, but that could be more clear. The Supplementary refers to block groups, but that doesn't appear anywhere in the main text. The Methods describes socio-demographic characteristics from the ACS, but the ACS doesn't report any measures at the block level as far as I am aware (I think the block group is the lowest level for the ACS, so the block data would come from the Census). It is not clear to me why the tract-county comparison (lines 116-123) are "localized" if the preceding section is about the block-county comparison, but maybe I'm misunderstanding the preceding section.

Thank you. We have clarified these points. We clarified the methods for Figure 1 and Figure 2 as aggregations.

We have also decided to remove the block group level part in the Supplemental Materials because it doesn't add meaningful additional information and may instead be confusing. We also added more language to our Methods describing the different levels of analysis more clearly.

We have added:

Risk of isolation is calculated at the census block level, which is the smallest geographical unit for which U.S. census data is available. To estimate populations at risk of isolation, we use population data from the 2020 U.S. Census (most recent). We combine the risk of isolation with socioeconomic and demographic data (including race, median income, median age, and percent of renting households) from the American Community Survey (ACS) to assess the spatial distribution of risk burden due to a disruption of transport connectivity.

The main text does not contain enough info on methods (I realize the methods come separately, but it's so minimal in the main text that I stopped reading, scrolled to the methods, and returned). I think a few more sentences would at least provide enough detail for the main text to stand on its own, while obviously readers more interested in the details can go to the methods section specifically. For example, the main text doesn't mention that the measure of isolation is done at the census block scale, using the block centroid.

Thank you for this comment. Other reviewers also pointed out that our methods were not clear enough in the main text. We have since added additional details in the main body of the text as well as clarified the methods where necessary in the Methods section. The main body now includes:

We argue that isolation creates a unique circumstance in which connectivity to essential

services has been disrupted on a highly localized spatial scale^{19,20}. In this paper, we measure risk using an established methodology that intersects OpenStreetMap (OSM) road network data with National Oceanographic and Atmospheric Administration (NOAA) mean higher high water (MHHW) scenarios for global SLR from 1 to 10 ft for all coastal counties in the continental U.S. Briefly, we compute the risk of isolation due to SLR by intersecting the U.S. OpenStreetMap (OSM) road network with NOAA's mean higher high water (MHHW) for global sea-level rise scenarios between 1 to 10 ft of global SLR at one-foot increments. Sea level will not rise uniformly. To account for this, we compute relative sea-level rise (RSLR) using tidal gauge data and SLR projections from Sweet et al. (2022). A census block is considered at risk of isolation if it lacks an available (not inundated) route between its centroid and any fire stations or primary schools at MHHW. We consider these services essential, and they also serve as a proxy for other key service, civic, and activity areas. The data, methods, and assumptions for calculating risk of isolation as well as a full discussion of limitations can be found in Logan et al. (2022).

For Figure 6 and associated results: the equation for the multivariate modeling is not included anywhere, and it was not clear to me what the units of SLR are and how it relates to the outcome variable – do you run the model for multiple levels of SLR and thus multiple outcome measures of inundation, isolation, etc.? The paper states that tracts with older, higher income, etc. are more likely to become isolated, but isn't it actually showing that a higher share of the tract is at risk of isolation, since the tract isn't being classified as isolated/not isolated?

The reviewer's explanation of the models in Figure 6 is correct. We have also added the equation for the regressions in the Methods.

We have clarified Figure 6 and the model (including adding a table, now Table 1 with the values of the coefficients for each model.

We have also added the clarifying text: "In these models, we include the level of SLR (1 to 10 ft) as an independent variable." We have also clarified in the discussion of the regression results that the results are indicating that a higher percent of the tract is at risk of isolation.

Second, there were a few major limitations that I did not see addressed anywhere in the text. Modeling isolation based on a census block centroid makes sense in urban areas where blocks are very small, but could be more substantially off for larger (in area) census blocks. Elevation can change substantially within some census blocks (thus much wider range of actual isolation) compared to others. The results assume that the entire population of the block is either isolated or not, and thus do not account for the within-block distribution of populations – it could be that socially vulnerable populations live in the most flood-prone parts of the block, so these results are an underestimate, or vice versa.

This is a good point. We have added the text below to our Discussion to address this and other major limitations.

This work has several important limitations. One limitation is that our analysis considers the entire population of a census block to be isolated based on transportation routes from the centroid of that census block. In this way, we are unable to consider finer-scale differences in isolation that may exist within a census block. It is possible that additional inequities in the risk of isolation due to SLR exist at an even finer spatial scale, which we are currently not able to capture. This work also does not consider possible future updates to road networks or other adaptive measures. Other limitations of the methodology for calculating risk of isolation are discussed in detail in Logan et al. 2022.

Minor comments:

- Overall: “risk of isolation,” “isolated,” “isolation risk exposure,” “effect of isolation due to SLR” are all used seemingly interchangeably throughout the paper. Is this intentional or are there subtle differences in the meanings? Suggest being consistent with how you describe what is being measured.

We intended to use these terms interchangeably but can see that this might be confusing. We have edited the text to use “risk of isolation” consistently.

- Abstract: specify the SLR amounts rather than “intermediate” which is subjective
We have clarified that intermediate means 4 ft or greater.

- Page 1, Lines 38-48 (and in some other parts of the paper): use of the terms hazard, risk, exposure, and vulnerability is inconsistent – need to either state definitions or be consistent with existing ones, e.g. IPCC (hazard, exposure, vulnerability as components of risk).
Thank you for pointing out that the use of these terms was not clear. We have removed reference to “hazard” other than in the literature review where we reference “natural hazards” broadly. We have also edited the lines that you point out specifically and generally try to avoid using “exposure”. As mentioned above, we have tried to consistently refer to our topic of interest as “risk of isolation”.

- Page 1, Lines 43-46: this is a modeled result using assumptions about economically optimal adaptation – this is not an empirical finding about likelihood of abandonment or protection.
We have clarified in this sentence that this is based on economic evaluation of adaptation and means that these areas “might be” more likely to be abandoned.

- Page 1, Lines 49-54: the distinction being drawn in this paper is not about inundation vs. some other metric of exposure, it’s about what is being inundated – a parcel, a % of land, vs. the roadways needed to access critical services.
Yes, that is a good point. We agree with this assessment. Part of our argument is that much planning, assessment of exposure, assessment of burden on communities relies on parcel inundation as the only metric. In reality, parcel inundation only captures some of the challenges associated with SLR (as you point out). Part of the difference with our metric

though is that the inundation of roadways is only a valuable metric when it results in isolation, or the disruption of access to essential services. Therefore, the isolation metric aims to more fully capture how SLR (based on what and where is inundated) will result in disruptions for communities. We have attempted to clarify that much of the current literature relies on “parcel inundation” rather than considering other infrastructure inundation and the implications of that inundation (i.e. isolation and lack of access to essential services).

- Figure 4: while there are some tracts where there are high ratios, it looks like in the majority of tracts, Hispanic and Black populations are not disproportionately at risk of isolation? I suggest avoiding cherry picking of specific results to describe here. Also flagging “disproportionate impact” as I don’t know if “impact” is the right term to use here, and it might be clearer to actually put the ratio in the axis label instead.

We have changed the axis label for clarity. The goal is not to cherry pick results here but to highlight that there are some tracts where Black and Hispanic populations are very overrepresented in the data and to point out that the distribution has a long tail. We have clarified this in the text:

“We find that the degree of disproportionate risk of isolation on racial minority groups is predicted to shift as SLR increases, consistent with results in Figure 1. While there is a wide distribution, we see that Black and Hispanic populations in some tracts experience a rate of isolation more than 10 times greater than their representation in the county population (Figure 4). This points to the highly uneven and localized nature of the risk of isolation on these racial groups.”

- Does Figure 5 only include tracts that are isolated or where at least one of the blocks/block groups within it are isolated?

We have clarified that Figure 5 includes all tracts.

- There were a few places where the language was confusing and hard to understand. For example, Lines 134-135: “...for those tracts that are predicted to disproportionately affect only...” it implies the tracts are affecting something. Similarly, Lines 107-108: “Both Black and Hispanic populations are estimated to result in disproportionate isolation...”

Thank you for pointing out these areas where the language is confusing. We have edited for clarity.

Reviewer #3 (Remarks to the Author):

The authors examine how an often-overlooked artifact of sea level rise, isolation, and its consequences will effect the accessibility to critical services due to inundation of transportation networks. The study utilizes several socio-demographic datasets in combination with sea level rise data from NOAA. Overall I think the manuscript will be of interest to planners but there are a couple clarifications/edits needed in regards to the sea level terminology and datasets.

1) Inundation vs. flooding needs to be clarified and consistently used throughout the manuscript. Inundation should be considered any need at or below mean higher high water (MHHW) which is the agreement (for now) defining typically dry from typically wet land often referred to as 'ground level' by NOAA National Weather Service.

We have clarified our text and improved the consistency of our wording. We have edited the text so that we clearly refer to “inundation” when considering whether or not a block is isolated due to SLR. In a few areas, we still use the term flooding but only to refer to other works.

2) Related, why should intersection of MHHW be considered the time horizon for 'isolation'. MHHW has a definition of being exceeded about every other day. Though this could be argued as an effective limit, where is the justification? This criteria needs to be introduced, discussed and better justified. One could argue isolation would occur prior to the intersection with MHHW (numerically approximated to about 180 days/year). How about 30 or 60 days per year? Perhaps the transportation network itself becomes compromised long before this MHHW intersection occurs. Again, if the goal is providing realistic future estimates of 'isolation' this criteria needs serious consideration. The authors could use the extreme water level probabilities to estimate future high water or flood frequencies to refine the isolation time horizons.

We appreciate the opportunity to clarify. Our study and preliminary literature (Logan et al., 2023) show that, for a given threshold, there is a risk of isolation above the risk of inundation. The main goal of this work is not to examine the exact value of that threshold, but rather, what is the relationship between isolation and inundation, in terms of the socio-demographic impacts and the distribution of those impacts concerning social equity. Changing the threshold will likely only affect the "when", which is not the primary purpose of this work.

We believe that the fields of geotechnical engineering and human behavior have yet to achieve concrete conclusions on what should be the “threshold” to evaluate the risk for either inundation or isolation. The authors believe that “When will water cover the road, on average, once per day”, as applied in this work, is less arbitrary than other depths. Mean Higher High Water (MHHW) measures the daily water level over 19 years, not the median value. The measurement hence does not rely on the “every other day” and “180 days/year” in our understanding.

We consider vehicle corrosion from repetitive exposure to salt water when the depth is as low as 1cm or simply the nuisance of needing to drive through water frequently. As the reviewer argued, which we agree, "the transportation network itself becomes compromised long before this MHHW intersection occurs", soil stability under many of these roads will likely weaken as a result of this tidal water exposure (and because of tidal action in some locations). However, the pavement and geotechnical research area is still in the developing stage. The authors refrain from engaging further in this discussion beyond the scope of analysis for this work.

3) The 1-10 foot contours on NOAA Sea Level Rise Viewer are not themselves scenarios of sea

level rise (SLR). The most recent scenarios are from the 2022 Interagency report (Sweet et al., 2022) which is appropriately referenced in the manuscript. The authors state that they use the SLR scenarios from the tide gauges, but the 2022 report also provided 1-degree gridded SLR scenarios which might better resolve spatially. If the goal is to be as realistic as possible, this gridded dataset should be used. Also, though the 2022 scenarios are defined as Low, Int Low, Intermediate, Int High and High, if 'more likely' estimates are sought, more emphasis should be given to the Int Low and Intermediate scenarios. The High is the upper end of plausible rise by 2100 and not very likely to occur.

We appreciate this comment, and we did elect to look at the Int Low scenario as well (0.5 ft of global SLR by 2100) in response to this comment. We see that, at this level of global SLR, disproportionate risk of isolation for Black and Hispanic communities are not evident when the data is aggregated to the entire U.S.. However, the point of this analysis is that the timing of national-level disproportionate risk of isolation depends highly on the global SLR scenarios. This is an important consideration for planning at that scale. But it does not mean that Black and Hispanic groups will not face disproportionate risk of isolation under the 0.5 ft, it just means that this will not be apparent at the national aggregated level.

We added a clarifying sentence to the Discussion that the timing will vary from place to place. We also chose to bring the timing information (based on the global SLR scenarios) into the main body of the paper from the Supplementary materials.

4) It is unclear how the time horizons for isolation under the Intermediate and High SLR scenarios were estimated and presented as one number for the country. It is important to attach time horizons to the SLR scenarios, not simply state, e.g. 'with 5' or 10' of sea level rise'. By when, where?

We have added details in the Methods section about how we took the scenarios from Sweet et al. 2022 and converted them into estimates from 2030 to 2100. We now write:

For timing of risk of isolation, we match each census block centroid to the nearest (by Euclidean distance) tidal gauge with relative sea-level projections from Sweet et al. 2022 climate scenarios and corresponding data22. We focus on the intermediate scenario (with a global mean sea level rise of 1.0 m by 2100) and high scenario (with a global mean sea level rise of 2.0 m by 2100). The intermediate scenario is considered the high end of likely SLR without rapid ice-sheet loss. The high projection represents the high end of SLR projections under high emissions. For both the intermediate and high scenarios, we consider the mean projections. A census block is considered isolated in a given 10-year interval (from 2030 to 2150) if the SLR projected at the nearest tidal gauge in Sweet et al. 2022 is equal to or greater than the SLR height needed to result in the isolation of that census block. The populations that are estimated to be at risk of isolation during a given 10-year interval are then summed across all census blocks included in the analysis.

REVIEWER COMMENTS

Reviewer #1 (Remarks to the Author):

The authors have addressed most of the comments on the previous draft and have greatly improved their description of their study methods. The results of the study show that, across the Continental United States (CONUS), Black and Hispanic residents will be disproportionately impacted by isolation from sea level rise once global mean sea level rise reaches ~ 4-6 feet. These results are not particularly surprising, given that Black and Hispanic residents disproportionately live in coastal states. It would have been interesting to also provide some context for the percentage of Black and Hispanic residents that live in coastal areas compared to the overall U.S.. The results of the study conducted at the census tract-level show that, while Blacks and Hispanic residents would be disproportionately burdened in a relatively small number of census tracts, in most tracts they're not.

There are several issues that should be addressed in the revised manuscript:

- The word 'communities' should not be used as a synonym for census tract.
- Terms like 'nationally' and 'overall U.S.' shouldn't be used as a synonym for CONUS-wide, since the study was limited to the CONUS.
- The methods section still includes the following text on lines 387 - 391:
"To further identify characteristics of communities susceptible to higher levels of isolation due to SLR as well as identify characteristics of populations that are not captured by analyses of inundation alone, we use our additional census data (i.e., age, disability, income, renters, and poverty status) along with isolation and inundation measurements for the multivariate regression analysis."

However, it seems like the disability and poverty status parameters were actually removed from the study in this version. If so, they should no longer be included in the study methods.

Reviewer #3 (Remarks to the Author):

The authors have appropriately addressed my original comments and no additional edits are required from my perspective. Nice work.

Reviewer #4 (Remarks to the Author):

Novel approach focusing on isolation as opposed to typical focus on inundation or economic risk. The approach is primarily technical, without delving into the implications of transportation route isolation for the populations assessed. As such, the primary significance appears to be methodological.

Research question 2 doesn't fully align with the focus of the analysis. The presentation of the RQ2 outcomes in the results isn't about varied characteristics but concurrent/conjoint disparities. Consider rephrasing RQ2.

Have you considered or assessed the effect of the census margin of error? The error in the estimates increases with decreasing spatial scale and tends to be higher for vulnerable populations. At the census block scale, the error margins are likely highest, and the resulting demographic estimates most unreliable. The exclusion of the error margin in the narrative is a notable omission, and I'm surprised it didn't appear in the limitations.

Is median household income even available at the block scale? I thought its smallest available aggregation was the block group.

Figures S12 and S23: Reduce the line weight on the county polygons so the color classification can be interpreted. The thick weight obscures the spatial distributions. For example, it isn't easy to visually discern clusters of counties referred to in line 145.

Lines 145-52 make an argument about the clustering of risk. How are you defining highly clustered? By visual inspection or by a more objective measure?

REVIEWER COMMENTS

Reviewer #1 (Remarks to the Author):

The authors have addressed most of the comments on the previous draft and have greatly improved their description of their study methods. The results of the study show that, across the Continental United States (CONUS), Black and Hispanic residents will be disproportionately impacted by isolation from sea level rise once global mean sea level rise reaches ~ 4-6 feet. These results are not particularly surprising, given that Black and Hispanic residents disproportionately live in coastal states. It would have been interesting to also provide some context for the percentage of Black and Hispanic residents that live in coastal areas compared to the overall U.S.. The results of the study conducted at the census tract-level show that, while Blacks and Hispanic residents would be disproportionately burdened in a relatively small number of census tracts, in most tracts they're not.

We thank the reviewer for their constructive feedback and the opportunity to improve this manuscript.

There are several issues that should be addressed in the revised manuscript:

- The word 'communities' should not be used as a synonym for census tract.

We have gone through the manuscript and changed "communities" to "census tracts" where needed.

- Terms like 'nationally' and 'overall U.S.' shouldn't be used as a synonym for CONUS-wide, since the study was limited to the CONUS.

We have corrected this. Please note that we use "overall U.S." in the figure captions for Figures 1 and 2, which is correct, as our baseline comparison is the national average.

- The methods section still includes the following text on lines 387 - 391:

"To further identify characteristics of communities susceptible to higher levels of isolation due to SLR as well as identify characteristics of populations that are not captured by analyses of inundation alone, we use our additional census data (i.e., age, disability, income, renters, and poverty status) along with isolation and inundation measurements for the multivariate regression analysis."

However, it seems like the disability and poverty status parameters were actually removed from the study in this version. If so, they should no longer be included in the study methods.

Thank you for helping us catch this. We have removed this from the Methods section and corrected the text to read:

To further identify characteristics of communities susceptible to higher levels of isolation due to SLR as well as identify characteristics of populations that are not captured by analyses of inundation alone, we use our additional census data (i.e., age, income, and renters) along with isolation and inundation measurements for the multivariate regression analysis.

Reviewer #3 (Remarks to the Author):

The authors have appropriately addressed my original comments and no additional edits are required from my perspective. Nice work.

We thank the reviewer for their time, comments, and encouragement.

Reviewer #4 (Remarks to the Author):

Novel approach focusing on isolation as opposed to typical focus on inundation or economic risk. The approach is primarily technical, without delving into the implications of transportation route isolation for the populations assessed. As such, the primary significance appears to be methodological.

Research question 2 doesn't fully align with the focus of the analysis. The presentation of the RQ2 outcomes in the results isn't about varied characteristics but concurrent/conjoint disparities. Consider rephrasing RQ2.

Thank you. We have revised RQ2 to be:

How does the risk of isolation correlate with socio-demographic characteristics associated with vulnerability (e.g., age, income, renter status, racial composition)?

Have you considered or assessed the effect of the census margin of error? The error in the estimates increases with decreasing spatial scale and tends to be higher for vulnerable populations. At the census block scale, the error margins are likely highest, and the resulting demographic estimates most unreliable. The exclusion of the error margin in the narrative is a notable omission, and I'm surprised it didn't appear in the limitations.

Thank you for raising this important point. We did not formally assess the margin of error in the census data. However, we have mentioned this now in the limitations of the work (pg 14):

There are also limitations in using census data for demographic information, as all census data has an associated margin of error. Therefore, there may be areas where populations, characteristics, and therefore risk of isolation are over- or under-estimated in this analysis.

Is median household income even available at the block scale? I thought its smallest available aggregation was the block group.

Yes, you are correct. We note that the analysis with median household income was done at the tract level for this reason.

Figures S12 and S23: Reduce the line weight on the county polygons so the color classification can be interpreted. The thick weight obscures the spatial distributions. For example, it isn't easy to visually discern clusters of counties referred to in line 145.

If the reviewer is referring to Figure S1 and Figure S2 in the supplementary materials, we have made the county polygon outlines thinner. We have also removed reference to “clustering” and instead focus on the broad spatial patterns (see below). Unfortunately, it is a little difficult to see county by county especially because there are some very small counties. We have tried to improve the visualization and explanation.

Lines 145-52 make an argument about the clustering of risk. How are you defining highly clustered? By visual inspection or by a more objective measure?

While we did do an objective clustering analysis (by counting the number of counties that have a disproportionate risk of isolation and also border at least one other county with the same disproportionate risk of isolation), we ultimately did not include it in this work. We have, therefore, removed the reference to clustering. The text now reads:

Black populations are disproportionately affected primarily in parts of the Northeast and California. We see counties where Hispanic populations are disproportionately affected across the study area, especially in Northern California, Louisiana, and large swaths of the Northeast (Figure 4).